# Relationship between Kinesiophobia, Foot Pain and Foot Function, and Disease Activity in Patients with Rheumatoid Arthritis: A Cross-Sectional Study

**DOI:** 10.3390/medicina59010147

**Published:** 2023-01-11

**Authors:** Andres Reinoso-Cobo, Ana Belen Ortega-Avila, Laura Ramos-Petersen, Jonatan García-Campos, George Banwell, Gabriel Gijon-Nogueron, Eva Lopezosa-Reca

**Affiliations:** 1Department of Nursing and Podiatry, Faculty of Health Sciences, University of Malaga, Arquitecto Francisco Peñalosa 3, Ampliación de Campus de Teatinos, 29071 Malaga, Spain; 2Instituto de Investigación Biomédica de Málaga (IBIMA), 29590 Malaga, Spain; 3Department of Behavioural Sciences and Health, University of Miguel Hernandez, 03202 Alicante, Spain; 4Institute for Health and Biomedical Research (ISABIAL), 03010 Alicante, Spain

**Keywords:** kinesiophobia, rheumatoid arthritis, foot, pain, function

## Abstract

The main objective of the present study was to determine the relationship between kinesiophobia and pain (general and foot pain), foot function, and disease activity in patients with rheumatoid arthritis (RA). A total of 124 interviews were carried out with participants with RA. Participants were recruited from the Hospital Universitario Virgen de las Nieves de Granada in Spain. Interviews took place in January 2021. Participants completed the following questionnaires during an appointment with their rheumatologist: Foot Function Index (FFI), Tampa Scale for Kinesiophobia (TSK-11), and the Visual Analogue Scale Pain foot (VAS Pain). Furthermore, the Simplified Disease Activity Index (SDAI) was used to measure disease activity. Of the 124 participants, 73% were women, and their mean age was 59.44 years (SD 11.26 years). In the statistical analysis, positive linear correlations (*p* < 0.001) were obtained between the variables TSK-11 and VAS (related to general pain or foot pain) and FFI (in its three subscales). Additionally, a negative correlation between the TSK-11 and the educational background was shown. This study provided information about the relationship between foot function and pain with different levels of kinesiophobia in patients with RA. Additionally, the educational background of the patient was considered a predictor of whether or not a patient suffered from kinesiophobia.

## 1. Introduction

Kinesiophobia is defined as the limitation of motor activity due to fear of movement. It was originally defined by Kori et al. in 1990, being considered as a weakening, destructive, and irrational fear of movement and activity based on the belief of being vulnerable and susceptible to injury [1]. This fear of movement causes self-defense; hence, patients avoid or limit the movements that require the action of the affected body segment [2].

Patients with rheumatoid arthritis (RA) suffer from pain in their joints, especially the small joints of the hands and feet [3]. RA causes pain, bone erosion, chronic synovitis, and increased elongation of the ligaments. These signs and symptoms cause higher deformity and greater stiffness [4]. During the course of the disease, 90% of the patients suffer from foot symptoms, which are often severe [5,6]. These symptoms are associated with an osteoarticular deformation [7] modifying the symmetry of the feet [8]. Therefore, patients with RA suffer from a detriment of their quality of life [9] and lower foot function [10].

In a high percentage of patients, even in the remission period of the disease, destruction of the joints has been observed. This leads to an alteration in joint alignment, instability, and a reduction in mobility [11]. Furthermore, foot symptoms associated with RA can cause biomechanical alterations, as well as a lack of joint proprioception. The level of physical activity may also be lower among individuals with RA when compared with healthy controls [10,12,13]. Consequently, patients with RA present a higher risk of falls, showing decreased postural stability and difficulty maintaining postural control during everyday activities [11].

Previous studies regarding kinesiophobia focus on the foot and ankle’s involvement, showing that people who present with ankle instability suffer from higher levels of kinesiophobia [14,15]. However, there are a lack of studies related to kinesiophobia and foot and ankle involvement in patients with RA. Some previous studies examined the level of fear of movement in patients with RA through the TSK-17 but on a general level [16]. From another perspective, some previous studies have been published regarding the relationship between kinesiophobia and the following areas: upper limbs in osteoarthritis [17,18,19], osteoporosis [20], and knee and hip surgery [21,22]. Some studies were focused on patients who suffer chronic pain [23], fibromyalgia [24,25], and musculoskeletal disorders [26]. Up until now, and as far as we know, no papers have been published that determine the predictive factors that are related to fear of movement in patients with RA.

The main objective of the present study was to determine the relationship between kinesiophobia and pain (general pain and foot pain), foot function, and disease activity.

## 2. Materials and Methods

### 2.1. Ethical Approval

The study has been approved by the Investigacion Medica de la Universidad de Malaga ethical committee (CEUMA-91-2015-H) and the Portal de Etica de la Investigacion Biomedica de Andalucía (PEIBA) (ARC001) in accordance with the Declaration of Helsinki; all methods were performed in accordance with the relevant guidelines and regulations.

### 2.2. Study Design

This study was a cross-sectional study.

### 2.3. Participants

Patients must have presented with foot pain and satisfy the 2010 RA classification criteria (approved by the American College of Rheumatology and the European League Against Rheumatism) [27]. 

Participants were excluded if they presented with concomitant musculoskeletal disease, central or peripheral nervous system disease, endocrine disorders (e.g., diabetes), or psychiatric disorders (e.g., depression).

Participants were recruited from the Hospital Universitario Virgen de las Nieves de Granada, Spain, from the rheumatology department. Participants were recruited from June 2021 to December 2021. They were provided with an information sheet and invited to participate in the study. The participants who agreed were interviewed and given further details about the study. All participants provided written informed consent to participate in the study.

### 2.4. Data Collection

The following demographic information was collected: years of age, gender, educational background, treatment, and disease duration. The following scales were used to collect clinical information to assess the disease activity and influence of the disease: the Visual Analogue Scale (VAS) [28] for both general pain (VAS-g) and foot pain (VAS-f), with a range of 0–100 where 0 is no pain and 100 is worst pain, and the Simplified Disease Activity Index (SDAI) to measure the disease activity for RA, which is a scoring system that has been validated in both research and clinical settings. Remission is defined as an SDAI of <3.3, low disease activity as ≤11, moderate disease activity as ≤26, and high disease activity as >26 [29]. The SDAI is the numerical sum of five outcome parameters: tender and swollen joint count (based on a 28-joint assessment), patient and physician global assessment of disease activity (VAS of 0–100 mm), and level of C-reactive protein (mg/dl, normal <1 mg/dl). Additionally, the Spanish version of the Tampa Scale for Kinesiophobia (TSK-11) [30] was collected to measure kinesiophobia. The TSK-11 is an 11-item self-report checklist using a 4-point Likert scale, with 1 being strongly disagree and 4 being strongly agree. Low scores mean no kinesiophobia and high scores indicate there is kinesiophobia. The TSK-11 presents good coherence (a 0.68–0.80), test–retest reliability (intraclass correlation coefficient [CCI] 0.72), and construct validity (the latter when tested against fear, pain, avoidance, and catastrophizing beliefs).

Finally, the Foot Function Index (FFI) was collected, which was developed to measure the impact of foot pathology on function in terms of pain, disability, and activity restriction. The FFI is a self-administered index consisting of 23 items divided into 3 sub-scales. Both total and sub-scale scores are produced. Higher FFI scores indicate poor foot health. Test–retest reliability of the FFI total and sub-scale scores ranged from 0.87 to 0.69. Internal consistency ranged from 0.96 to 0.73 [31].

### 2.5. Procedure

Two independent researchers interviewed the participants face to face. The interview took place in a hospital room, where the participants were also asked to complete the questionnaires described previously and some demographic questions. Additionally, the participants weight and height were assessed.

### 2.6. Sample Size Calculation

The sample size was determined with EPIDAT 4.1, calculating an effect size of 0.61. This was calculated from a pilot study to evaluate the statistical power and used the TSK-11 as the primary variable. Finally, the pilot study involved a sample of 30 subjects. The study was designed to detect changes exceeding 0.8 (high effect size) for a variation in the sample according to the above classification, with a type I error of 0.05 and a type II error of 0.2. The result of sample size calculation was that at least 114 subjects were required, but 124 participants were included.

### 2.7. Statistical Analysis

The results obtained are reported as the mean and standard deviation (SD) due to the normal distribution of the sample. The normality of the distributions was examined by the Kolmogorov–Smirnov test. Pearson’s correlations were used for inter-item and item-total correlations in view of the normal distribution observed in most cases. Finally, a multivariable linear regression model was obtained to evaluate the predictors of kinesiophobia, according to the Spanish version of the TSK-11. In constructing the models, the regression assumptions of homoscedasticity, normality, and independence of the residuals and collinearity were tested. Homoscedasticity was evaluated by analyzing the distribution of predicted values and scatterplots of the residuals. Normality of the residuals was tested by analyzing histograms and by graphs of standardized residuals. Independence of the residuals was evaluated by the Durbin–Watson statistics. Finally, the presence of collinearity was tested by calculating the variance inflation factor, the tolerance, and partial correlations. The level of significance was established at *p* < 0.05. All the statistical analyses were assessed using SPSS software (IBM SPSS Statistics: Version 25, Chicago, IL, USA).

## 3. Results

In total, 147 participants were recruited, but five were not enrolled because they did not provide informed written consent. A total of 12 participants were excluded because they did not complete the questionnaires, and six participants declined to be enrolled in the study. Finally, 124 participants were included in the study and analyzed (average duration of RA 18.86 years, SD 10.61 years), and 91 patients (73.4%) were female. The mean age was 59.44 years (SD 11.26). The mean height and weight were 163.31 cm (SD 12.06) and 71.1 kg (SD 15.5), respectively, and the Body Mass Index (BMI) was 26.83 (SD 6.65) (Table 1). The patients with RA were treated with biological disease-modifying antirheumatic drugs (DMARDs) (42%), methotrexate (35%), or non-steroidal anti-inflammatory drugs/corticosteroids (20%). The educational background of the participants was no formal education 9 (7.3%), primary school 59 (47.6%), high school 22 (17.7%), and higher education 34 (27.4%).

The results in pain (VAS-g) and disability (FFI) correlate with a higher level of kinesiophobia of 31.77 (7.77). Pain measured by the VAS-g was 5.88 (SD 3.18), and disability measured by the FFI was 40.56 (SD 29.23). However, an association was not shown between patients who are in remission of the disease and FFI. A total of 22.8% of the patients were in disease activity remission measured by the SDAI, 38.7% of them presented low levels of disease activity, 32.3% of them presented moderate levels of disease activity, and 6.5% of them presented high levels of disease activity (Table 1).

The TSK-11 is highly correlated with pain levels, both general and foot pain (VAS-g 0.504 and VAS-f 0.421) (Table 2). The results show a correlation between the education background and the level of kinesiophobia, showing that patients with a high education background presented a lower level of kinesiophobia (*p* < 0.001 and eta squared (η2) 0.197).

There is a positive correlation except for the BMI, which means that high levels of kinesiophobia indicate high levels of foot pain and general pain: r of 0.421 ** and 0.504 **, respectively. However, there is no strong correlation between FFI and kinesiophobia, where high levels of kinesiophobia indicated low levels of FFI in all its domains. Finally, low BMI indicated high levels of kinesiophobia (Table 2).

According to the data for the TSK-11, the multivariable linear regression presented an R2 value of 34.1%. (Table 3). Post-hoc analysis yielded a power of 0.95 for this four-predictor model. There was no collinearity in the model (maximum VIF 2.15 and minimum tolerance of 0.6), and the residuals were independent (Durbin–Watson, 2.11) when using the same parameters (i.e., age, foot VAS pain, BMI, activity level of SDAI and FFI physical activity).

## 4. Discussion

The main objective of the present study was to determine the relationship between kinesiophobia and pain (general and foot pain), foot function, and disease activity.

Our findings show the relationship between foot function and pain with high levels of kinesiophobia in patients with RA. Patients showed a very high level of kinesiophobia (TSK-11 = 31.77) and both general pain and specific foot pain. Furthermore, patients presented low foot function (FFI = 39.72), meaning that there is an association between the included outcomes. 

Previous studies agree with the present results, such as Kinikli et al., where pain and function were considered as predictive factors to suffer from kinesiophobia [16]. Additionally, Luque et al., Bilgin et al. and Luijsterburg et al. [26,32,33] demonstrated an association between kinesiophobia and musculoskeletal pain (mainly in the shoulder, neck, and back). All the authors suggested that the intensity of pain may cause the avoidance of performing movements that could increase pain. 

The previous studies analyzed pain in the upper limbs, as opposed to the present study, which analyzes both general and specific foot-related pain, which is one of the most affected areas in patients with RA. It has been shown in previous studies focused on RA that patients suffer from pain due to the disease and to structural deformities [7,8], provoking a physical detriment in patients with RA [9]. It should be pointed out that feet are one of the most important areas of the human body due to their role in the locomotor system [34] and, therefore, in the performance of physical and other activities, meaning that if patients present with foot pain, their activities are going to be reduced [6]. This statement has been confirmed by Palomo Lopez et al., where the level of kinesiophobia was studied in healthy participants who presented with hallux valgus (TSK-11 = 26.36) [15], and by Megan N. et al., which included patients with chronic ankle instability [12]. Crombez et al. demonstrated that acute musculoskeletal pain may cause chronic pain provoking disability in patients with high levels of kinesiophobia [35]. Naugle et al. suggest that greater pain-related fear of movement/injury is associated with lower levels of physical activity, greater sedentary behavior, and poorer physical function in adults. These results show the potential negative impact of kinesiophobia in older adults who do not report chronic pain [36].

However, in the presented results, kinesiophobia was not directly related with disease activity. Our results were supported by a previous study by Kinikli et al., where a statistically significant relationship was not found between RA disease activity, analyzed with the DAS28, and physical activity levels. Additionally, it was concluded that physical activity levels were not predictive of kinesiophobia [16]. 

The present study shows a significant and inverse relationship between the educational background of patients with RA and the level of kinesiophobia. Highly educated patients presented with a low level of kinesiophobia. This fact could be explained because highly educated patients were able to understand the provided information [37], such as the provided guidelines related to the conservative treatment related to their disease. 

Additionally, unlike patients with a low level of education, highly educated patients have access to tools to help in their pain management, such as databases of scientific papers such as PubMed, where they can read about the most recent recommendations for their disease, improving their lives and their pain levels. Furthermore, social activities may have an influence as well [38]. In the same way, it has been illustrated that pain and kinesiophobia outcomes are related to socio-economic variables [39], suffering from depression and/or anxiety, and the educational background of the patient [32].

Patients with high levels of kinesiophobia present greater functional problems in the foot and lower extremities [14,40]. In the case of hallux valgus, moderate or severe deformities are associated with higher levels of kinesiophobia and pain [15]. 

Finally, it is important to emphasize that kinesiophobia has been related to other outcomes such as quality of life [41,42], disability [41,43], and intensity and severity of pain [44,45]; however, there was no homogeneity in terms of the scales that were implemented.

This research presents some limitations. First, it is a cross-sectional study, which means that researchers were not able to detect developments or changes in the characteristics of the participants. Additionally, a convenience sampling was recruited, which made an impact on the homogeneity of the gender of the patients. Regarding the outcomes that measure foot pain, the results could be altered and increased because the sample was composed of mainly women. For example, one external factor typical in women may be footwear choice, and even though such data were not collected in the present manuscript, previous studies have highlighted this issue [46,47]. Foot deformities greatly impacts footwear choice, as poorly fitting footwear can cause pain, which can lead to lack of movement and function. Additionally, the number of years that patients have been taking medication should have been considered, as well as the type of medication, which may be related to the pain perception and kinesiophobia. In addition, the TSK-11 has not been validated in patients with RA. Finally, the mean age could also be considered as a limitation of the present study. The mean age (59.44 years) could have biased the results due to older patients supposedly presenting concomitant pathologies that have not been considered, and this may have influenced the results in terms of kinesiophobia.

The strengths of this study include: the analysis of the relationship between general pain, foot pain, and foot function with kinesiophobia in patients with RA, which is a disease that requires a lot of resources from the healthcare system, including expensive treatments; it is a study that does not show a relationship between the disease activity and kinesiophobia, which agrees with previous studies that have been mentioned in the present discussion section. There is a relationship between pain, function, and kinesiophobia and also between limitation of physical activity and kinesiophobia. It may be hypothesized that patients with RA who increase their physical activity may suffer from higher levels of pain, but due to their level of disease activity and not their level of movement. 

It has been demonstrated that patients who suffer from musculoskeletal pain in general or foot pain in particular present high levels of kinesiophobia [40,48]. Therefore, the most suitable conservative treatments for musculoskeletal problems, including supervised physical activity and orthopedic treatments (i.e., foot orthoses), can be recommended knowing the level of kinesiophobia. This more targeted conservative treatment can help improve the patients’ range of movement. Furthermore, a better quality of life would be ensured on a psychological and physical level, which would reduce the treatment costs and improve the perception of pain.

There are some recommendations to guide future research, such as increasing the sample size to ensure homogeneity in terms of the following: the patients’ gender, the educational background of the patients, the techniques used to manage the pain, and the use of conservative treatments prescribed by healthcare professionals with the purpose of improving pain perception and kinesiophobia. Future research should also include the psychosocial and emotional perception of the patients to study their quality of life in the long term. In addition, this study could have collected the selected questionnaires as part of the rheumatology appointment, making the sample size more representative; information about targeted therapies such as foot orthoses or similar podiatric interventions could have also been collected to examine if there is a significant statistical relationship between kinesiophobia and patients who wear podiatric treatments or not.

## 5. Conclusions

Disease activity, general VAS, foot VAS, FFI physical activity, Body Mass Index, and age are all related to the level of kinesiophobia in patients with RA.

This study provided information regarding the relationship between poor foot function and high levels of foot pain with high levels of kinesiophobia in patients with RA. Additionally, a lower level of education was considered a predictor of kinesiophobia.

## Figures and Tables

**Table 1 medicina-59-00147-t001:** Characteristics of the sample reported as the mean and standard deviation (SD) and confidence interval.

	Mean (SD)	95% CI	
Age (years)	59.44 (11.26)	57.38	61.51
Disease duration (years)	18.86 (10.61)	16.92	20.81
Height (cm)	163.31 (12.06)	161.1	165.52
Weight (kg)	71.1 (15.50)	68.26	73.94
VAS-g	5.88 (3.18)	5.3	6.46
VAS-f	5.84 (3.44)	5.21	6.47
TSK-11_Total	31.77 (7.77)	30.35	33.19
Body Mass Index (kg/m^2^)	26.83 (6.65)	25.65	26.25
SDAI	10.73 (8.43)	9.19	12.27
FFI Total	39.72 (27.23)	34.74	44.71
FFI Pain	39.74 (25.38)	35.09	44.38
FFI Disability	40.56 (29.23)	35.21	45.92
FFI Physical_Activity	11.07 (12.43)	8.79	13.34

VAS-g: Visual Analogue Scale general pain; VAS-f: Visual Analogue Scale foot pain; TSK: Tampa Scale for Kinesiophobia; SDAI: Simplified Disease Activity Index; FFI: Foot Function Index; SD: standard deviation; CI: confidence interval.

**Table 2 medicina-59-00147-t002:** TSK-11 Total bivariant correlation by Pearson correlation.

N = 124	Pearson Correlation	*p* Value
Age (years)	0.262 **	0.003
Disease duration (years)	0.102	0.259
VAS-g	0.421 **	<0.001
VAS-f	0.504 **	<0.001
FFI Total	0.368 **	<0.001
FFI Pain	0.311 **	<0.001
FFI Disability	0.367 **	<0.001
FFI Physical Activity	0.250 **	0.005
BMI	−0.282 **	<0.001
SDAI	0.162	0.073

VAS-g: Visual Analogue Scale general pain; VAS-f: Visual Analogue Scale foot pain; FFI: Foot Function Index; BMI: Body Mass Index; SDAI: Simplified Disease Activity Index.* The correlation is significant at the 0.05 level (bilateral). ** The correlation is significant at the 0.01 level (bilateral).

**Table 3 medicina-59-00147-t003:** Model of multivariable regression for the association of kinesiophobia.

				95%CI	
	B	β	*p*	Lower	Upper
SDAI	0.098	0.106	0.179	−0.046	0.241
VASf	0.682	0.298	0.007	0.193	1.17
VASg	0.467	0.192	0.058	−0.017	0.951
FFI Physical Activity	−0.005	−0.009	0.92	−0.113	0.102
Body Mass Index	0.222	0.191	0.017	0.04	0.403
Age	0.079	0.116	0.149	−0.029	0.187
	0.148	0.127	0.111	−0.035	0.33

VAS-g: Visual Analogue Scale general pain; VAS-f: Visual Analogue Scale foot pain; FFI: Foot Function Index; SDAI: Simplified Disease Activity Index.

## Data Availability

Not applicable.

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
