# Peer review of "Relationship between Kinesiophobia, Foot Pain and Foot Function, and Disease Activity in Patients with Rheumatoid Arthritis: A Cross-Sectional Study"

_medicina, 2023, doi:10.3390/medicina59010147_

Round 1
Reviewer 1 Report (New Reviewer)
Dear Authors,
Thank you for submitting this manuscript for publication. Overall, you have presented the paper excellently and the content was easy to follow and showed sound scientific writing.
May you please see the following comments/feedback I have provided that I hope will aim to improve this paper for prior to acceptance for publication:
Introduction
Lines 38-39 – “Patients with rheumatoid arthritis (RA) suffer from pain due to certain joint movements, especially the small joints of the hands and feet” – Sorry this statement somewhat confuses me. RA patients suffer pain in joints, particularly the hands and feet but their pain is not due to movement per se. I would say their pain is due more because of synovitis and then it hurts to move due to inflammation, stiffness etc. I would suggest rewording this statement to say something like “Patients with RA suffer from pain in joints, especially the small joints of the hands and feet”. You can use this statement to then lead you into the next sentence.
The remaining information for the introduction is very well presented. Well done.
Methods
Is there a specific reason why height and weight was measured and presented in this paper?
Has the TSK-11 been validated in patients with rheumatoid arthritis? If not I think this should be stated clearly and listed as a limitation.
Foot Function Index – Can you add some information here on whether a higher or lower FFI score is a more desirable outcome so the reader can interpret the results more clearly. I would also recommend doing this for the TSK-11.
Regarding the inclusion and exclusion criteria, did the authors consider those patients who have or haven’t recently received any specific or targeted therapies such as foot orthoses or other similar podiatric interventions? Or was this collected at all as part of the participants characteristics? I feel this is probably an important factor to consider given this paper is asking about RA patients and foot problems. If the authors have not collected this data I would suggest highlighting this as a limitation and possible need to capture this in future studies.
Results
One thing I have noticed in your results section is that you state that two variables are highly correlated or correlated. It is unclear whether you mean positively or negatively correlated and as such I would highly recommend you doing this so the reader can understand what direction the variables are going in relation to TSK-11. This is highly important as you report that pain generally and foot are highly correlated with TSK-11. Are they highly negatively or positively correlated? Your cohort of participants display a very LOW average of pain in both categories and so I suspect that the your conclusion should be the lower the pain the higher the TSK-11 and therefore kinesiophobia? It would seem odd to conclude that a higher amount of pain results in higher kinesiophobia based on the results of this study when your pain outcomes are so low? Some clarification on this is needed please.
Discussion
Lines 232-233 “highly educated patients have 232 access to tools to help in their pain management”. What sort of tools? Could you be more specific here please.
Line 240 “Kinesiophobia levels are associated to foot function (14)”. I would remove this statement as it doesn’t add anything and is superseded by the next statement.
Line 253-254 Regarding your statement with women choosing poorer footwear. Did you collect footwear data in your study? If not then I would recommend removing this comment or rephrasing to say something like “It is unclear if footwear choice was a confounding factor in this study as this data was not collected….”
Lines 271-273 Are there any appropriate references for this information? If not I would recommend stating there is little to no evidence before making recommendations on clinical practice.
Line 274 “orthopedic treatments” what do you mean by orthopedic treatments? May you please be more specific or provide examples
Conclusion
What type of relationship does foot function and foot pain have on levels of Kinesiophobia in patients with RA? Please be more specific.
Author Response
Dear Authors,
Thank you for submitting this manuscript for publication. Overall, you have presented the paper excellently and the content was easy to follow and showed sound scientific writing.
May you please see the following comments/feedback I have provided that I hope will aim to improve this paper for prior to acceptance for publication:
R. Dear reviewer,
Thank you very much for giving us the possibility of addressing all the questions that arose during the review process. We think those comments have greatly improved the quality of this cross sectional study. Please find below all the responses in a point-by-point fashion. In the new revised version, the changes are highlighted in red.
Introduction
Lines 38-39 – “Patients with rheumatoid arthritis (RA) suffer from pain due to certain joint movements, especially the small joints of the hands and feet” – Sorry this statement somewhat confuses me. RA patients suffer pain in joints, particularly the hands and feet but their pain is not due to movement per se. I would say their pain is due more because of synovitis and then it hurts to move due to inflammation, stiffness etc. I would suggest rewording this statement to say something like “Patients with RA suffer from pain in joints, especially the small joints of the hands and feet”. You can use this statement to then lead you into the next sentence.
R. Thank you for the suggestion. The sentence has been changed as suggested.
The remaining information for the introduction is very well presented. Well done.
Methods
Is there a specific reason why height and weight was measured and presented in this paper?
R. Height and weight were measured as part of the characteristics of the sample, to describe the participants.
Has the TSK-11 been validated in patients with rheumatoid arthritis? If not I think this should be stated clearly and listed as a limitation.
R. The TSK-11 has not been validated for patients with RA, and that information has been added as a limitation now.
Foot Function Index – Can you add some information here on whether a higher or lower FFI score is a more desirable outcome so the reader can interpret the results more clearly. I would also recommend doing this for the TSK-11.
R. The following information has been added: “Higher FFI scores indicate poor foot health”. The description for the TSK-11 is available in the manuscript “Low scores mean no kinesiophobia and high scores indicate there is kinesiophobia”.
Regarding the inclusion and exclusion criteria, did the authors consider those patients who have or haven’t recently received any specific or targeted therapies such as foot orthoses or other similar podiatric interventions? Or was this collected at all as part of the participants characteristics? I feel this is probably an important factor to consider given this paper is asking about RA patients and foot problems. If the authors have not collected this data I would suggest highlighting this as a limitation and possible need to capture this in future studies.
R. That information has not been collected. In the reviewed manuscript, we have suggested collecting targeted therapies in future studies: “Finally, collecting information about targeted therapies such as foot orthoses or similar podiatric interventions, to examine if there is a significant statistic relationship between kinesiophobia and patients who wear podiatric treatments or not.”
Results
One thing I have noticed in your results section is that you state that two variables are highly correlated or correlated. It is unclear whether you mean positively or negatively correlated and as such I would highly recommend you doing this so the reader can understand what direction the variables are going in relation to TSK-11. This is highly important as you report that pain generally and foot are highly correlated with TSK-11. Are they highly negatively or positively correlated? Your cohort of participants display a very LOW average of pain in both categories and so I suspect that the your conclusion should be the lower the pain the higher the TSK-11 and therefore kinesiophobia? It would seem odd to conclude that a higher amount of pain results in higher kinesiophobia based on the results of this study when your pain outcomes are so low? Some clarification on this is needed please.
R. There is a positive correlation except for the BMI, which means that high levels of kinesiophobia, indicated high levels of foot pain and general pain: r of 0.421** and 0.504**. However, there is no strong correlation between FFI and kinesiophobia, where high levels of kinesiophobia, indicated low levels of FFI in all its domains. Finally, low BMI indicated high levels of kinesiophobia (Table 2).
That information has been clarified in the manuscript.
Discussion
Lines 232-233 “highly educated patients have 232 access to tools to help in their pain management”. What sort of tools? Could you be more specific here please.
R. The following information has been added: “such as databases of scientific papers like Pubmed where they can read about the most recent recommendations for their disease”.
Line 240 “Kinesiophobia levels are associated to foot function (14)”. I would remove this statement as it doesn’t add anything and is superseded by the next statement.
R. The sentence has been removed.
Line 253-254 Regarding your statement with women choosing poorer footwear. Did you collect footwear data in your study? If not then I would recommend removing this comment or rephrasing to say something like “It is unclear if footwear choice was a confounding factor in this study as this data was not collected….”
R. The sentence has been rephrased: “For example, one external factor typical in women may be footwear choice, and even though that data was not collected in the present manuscript, previous studies have highlighted the issue”. Also, a new reference has been added: Williams AE, Nester CJ, Ravey MI. Rheumatoid arthritis patients' experiences of wearing therapeutic footwear - a qualitative investigation. BMC Musculoskelet Disord. 2007 Nov 1;8:104. doi: 10.1186/1471-2474-8-104. PMID: 17976235; PMCID: PMC2190761.
Lines 271-273 Are there any appropriate references for this information? If not I would recommend stating there is little to no evidence before making recommendations on clinical practice.
R. The following references have been added:
Cotchett M, Frescos N, Whittaker GA, Bonanno DR. Psychological factors associated with foot and ankle pain: a mixed methods systematic review. J Foot Ankle Res. 2022 Feb 3;15(1):10. doi: 10.1186/s13047-021-00506-3. PMID: 35115024; PMCID: PMC8812226.
Yildiz S, Kirdi E, Bek N. Comparison of the lower extremity function of patients with foot problems according to the level of kinesiophobia. Somatosens Mot Res. 2020 Dec;37(4):284-287. doi: 10.1080/08990220.2020.1823362. Epub 2020 Sep 30. PMID: 32996810.
Line 274 “orthopedic treatments” what do you mean by orthopedic treatments? May you please be more specific or provide examples
R. The following information has been added: “(i.e., foot orthoses)”.
Conclusion
What type of relationship does foot function and foot pain have on levels of Kinesiophobia in patients with RA? Please be more specific.
R. The following information has been added: “This study provided information about the relationship between poor foot function and high levels of foot pain with high levels of kinesiophobia in patients with RA”.

Reviewer 2 Report (New Reviewer)
The study demonstrates the psychological and social aspects of a chronic painful condition, such as rheumatoid arthritis. A fear of use and function of someone’s body does directly impact their quality of life and overall economic by ability. The nature of the study being of a somewhat volunteer survey does limit its overall objectiveness, but still provides significant value. It is recommended that more detail, in the discussion section, should include the nature of this study design as well as further improvements to a study that could be done at this nature.
Author Response
Dear reviewer,
Thank you very much for giving us the possibility of addressing all the questions that arose during the review process. We think those comments have greatly improved the quality of this cross sectional study. Please find below all the responses in a point-by-point fashion. In the new revised version, the changes are highlighted in red.
We carried out a non-probabilistic convenience sampling using the "consecutive sampling" technique, where the participants who met the previously specified selection criteria were selected. This technique is used when, at the beginning of the study, there is no list of the population that will develop the disease and, as it was the case in our study, when it is unknown which patient is going to have pain. In those cases, a simple random sampling could not be carried out.
Argimon Pallás, J. M., & Jiménez Villa, J. (2000). Métodos de investigación: clínica y epidemiológica.
The following information has been added: “In addition, to collect the selected questionnaires for this study as part of the rheumatology appointment, so the sample size will be more representative”.

Round 2
Reviewer 1 Report (New Reviewer)
All comments have been addressed.
Thank you and all my best,
Antoni
Author Response
Dear reviewer,
Thank you very much for giving us the possibility of addressing all the questions that arose during the review process. We think those comments have greatly improved the quality of this cross sectional study.
The following sentence has been added in the mentioned paragraph: “…We hypothesise that this fact could be explained because highly educated patients…”.

This manuscript is a resubmission of an earlier submission. The following is a list of the peer review reports and author responses from that submission.
Round 1
Reviewer 1 Report
General considerations: In this manuscript Reinoso-Cobo and cols. studied the relation between kinesiophobia (Tampa Scale for Kinesiophobia –TKS-11) and pain (general pain, foot pain), foot function (Foot Function Index), disease activity (SDAI) and some clinical data (age, disease duration, BMI, education level). There was significant TKS-11 correlation with pain, foot function, age and education; but not with disease duration and disease activity.
The study proposal is original and deserves attention. However, the study needs to be revised, especially in relation to the description of the results and the discussion.
Abstract: line 6 – remove “in patients with rheumatoid arthritis” (repeated)
Introduction: You say that the objective of the study was to determine the factors related to teh development of kinesiophobia. I think this is not your real objective. I recommend you to keep only determine the relation between kinesiophobia and pain (general and foot pain), foot function and disease activity.
Methods: In the item 2.3 Participants it is recommended to show the eligibility criteria, and the sources and methods of selection of participants. Report numbers of individuals at each stage of study only in Results.
Line 97-99: regarding visual analogue pain, please describe that you will collect both general pain and feet pain in separate, as it is shown in results. I recommend you to use a single nomenclature/abbreviation for general pain and foot pain throughout the text (for ex. Sometimes you use VAS in general as in Table 1, and then VAS in general, as in Table 2)
Results:
The description of data in the paragraphs that explain Tables 1 an 2 Lines 160 – 171 is really very confuse, difficult to understand.
I recommend you to let any discussion or personal opinion (such as you did in line 169 “however, it is important to indicate…”) to discussion. .
Tables – For each table include in the title or in footnotes the details of the type of statistics presented or the analytical method.
I recommend you to change: years of age (years) to Age (years); disease evolution to Disease duration; please give proper nomenclature and abbreviations for pain general VAS and foot pain VAS – as cited before.
In relation to table 1 please it would be important to describe the strength of association between variables and TSK-11 (small, medium or large).
Beyond Pearson correlation, did you compared TKS between the groups of patients with low, moderate and high disease activity according to SDAI?
Discussion
Line 210 Patients with RA who suffer from foot pain associated to high levels of kinesiophobia, showed dissatisfaction and even depression. – reference is missing.
Instead of “the results included in the present study, prove that kinesiophobia is not directly related with the disease activity” , it would be better: “in this study/ In the presented results kinesiophobia was not directly related with the disease activity” (line 214)
Many of your references do not include kinesiophobia, but pain perception after surgery (37), pain, function and pain catastrophizing (38, 39). It is recommended that you keep this clear in the text.
It was important to notice that TSK-11 did not relate with SCDAI (disease activity), but with pain and education. Unfortunatelly this can be better discussed.
Conclusion
You said: “…SDAI and Body Mass Index are all related to a high level of kinesiophobia” and in discussion “ … A clear and significant tendency was not shown between a greater or lower level of kinesiophobia and the disease activity”. This is contradictory. Please review.
Author Response
General considerations: In this manuscript Reinoso-Cobo and cols. studied the relation between kinesiophobia (Tampa Scale for Kinesiophobia –TKS-11) and pain (general pain, foot pain), foot function (Foot Function Index), disease activity (SDAI) and some clinical data (age, disease duration, BMI, education level). There was significant TKS-11 correlation with pain, foot function, age and education; but not with disease duration and disease activity.
The study proposal is original and deserves attention. However, the study needs to be revised, especially in relation to the description of the results and the discussion.
R. Dear reviewer,
Thank you very much for giving us the possibility of addressing all the questions that arose during the review process. We think those comments have greatly improved the quality of this cross-sectional study. Please find below all the responses in a point-by-point fashion. In the new revised version, the changes are highlighted in red font.
Abstract: line 6 – remove “in patients with rheumatoid arthritis” (repeated)
R. Thank you, this duplicated phrase has been deleted from the manuscript.
Introduction: You say that the objective of the study was to determine the factors related to teh development of kinesiophobia. I think this is not your real objective. I recommend you to keep only determine the relation between kinesiophobia and pain (general and foot pain), foot function and disease activity.
R. The objective has been modified: “The main objective of the present study was to determine the relationship between kinesiophobia and pain (general and foot pain), foot function and disease activity.”
Methods: In the item 2.3 Participants it is recommended to show the eligibility criteria, and the sources and methods of selection of participants. Report numbers of individuals at each stage of study only in Results.
R. The participants results have been deleted from the methods section and they are now available in the results section.
Line 97-99: regarding visual analogue pain, please describe that you will collect both general pain and feet pain in separate, as it is shown in results. I recommend you to use a single nomenclature/abbreviation for general pain and foot pain throughout the text (for ex. Sometimes you use VAS in general as in Table 1, and then VAS in general, as in Table 2)
R. The following sentence has been added to the manuscript, in the data collection section: “…the Visual Analogue Scale (VAS) for pain [28] for both general pain (VAS-g) and feet pain (VAS-f)…” Also, the terms VAS-g and VAS-f are now specified in table 1, 2 and 3.
Results:
The description of data in the paragraphs that explain Tables 1 an 2 Lines 160 – 171 is really very confuse, difficult to understand.
R. The sentence has been rewritten: “The results in pain (VAS-g) and disability (FFI) correlate with a higher level of kinesiophobia, being 31.77 (7.77). Pain measured by the VAS-g was 5.88 (SD 3.18) and disability measured by the FFI was 40.56 (SD 29.23). However, an association was not shown between patients who are in remission of the disease and FFI. 22.8% of the patients were in disease activity remission measured by SDAI, 38.7% of them presented low levels of dis-ease activity, 32.3% of them moderate levels of disease activity and 6.5% of them high levels of disease activity (table 1).
The TSK-11 is highly correlated with pain levels, both in general and foot pain (VAS-g 0.504 and VAS-f 0.421) (table 2). The results by ANOVA one-way also show a correlation between the education background and the level of kinesiophobia, showing that patients with a high education background presented lower level of kinesiophobia (p<0.001 and eta squared (η2) 0.197).”
I recommend you to let any discussion or personal opinion (such as you did in line 169 “however, it is important to indicate…”) to discussion. .
R. This phrase has been rewritten: “The results by ANOVA one-way also show a correlation between the education back-ground and the level of kinesiophobia, showing that patients with a high education back-ground presented lower level of kinesiophobia (p<0.001 and eta squared (η2) 0.197).”
Tables – For each table include in the title or in footnotes the details of the type of statistics presented or the analytical method.
R. The following sentences have been added after each table specifying the type of statistics:
- “Table 1. Characteristics of the sample by the mean and standard deviation (SD) and confidence interval.”
- “Table 2. TSK-11 Total bivariant correlation by Pearson correlation.”
- “Table 3. Model of multivariable regression for the associated of Kinesiophobia.”
I recommend you to change: years of age (years) to Age (years); disease evolution to Disease duration; please give proper nomenclature and abbreviations for pain general VAS and foot pain VAS – as cited before.
R. Thank you. The terms have been changed and the nomenclatures have been added.
In relation to table 1 please it would be important to describe the strength of association between variables and TSK-11 (small, medium or large).
R. We are sorry, but TSK-11 does not permit different categories (small, medium or large), due to it is a continuous variable.
Beyond Pearson correlation, did you compared TKS between the groups of patients with low, moderate and high disease activity according to SDAI?
R. An association was not shown between patients who are in remission of the disease and FFI. 22.8% of the patients were in disease activity remission measured by SDAI, 38.7% of them presented low levels of disease activity, 32.3% of them moderate levels of disease activity and 6.5% of them high levels of disease activity (table 1).
Discussion
Line 210 Patients with RA who suffer from foot pain associated to high levels of kinesiophobia, showed dissatisfaction and even depression. – reference is missing.
R. Thank you. We have decided to delete that part of the discussion, avoiding mentioning “depression” as it is not an outcome that we have included in our study.
Instead of “the results included in the present study, prove that kinesiophobia is not directly related with the disease activity” , it would be better: “in this study/ In the presented results kinesiophobia was not directly related with the disease activity” (line 214)
R. The sentence has been modified as suggested and we have added the following information: “However, in the presented results kinesiophobia was not directly related with the disease activity. Our results were supported by the previous study by Kinikli et al., where a statistically significant relationship was not found between RA disease activity, analyzed with the DAS28, and physical activity levels. Also, it was concluded that physical activity levels were not predictive of kinesiophobia (16).”
Many of your references do not include kinesiophobia, but pain perception after surgery (37), pain, function and pain catastrophizing (38, 39). It is recommended that you keep this clear in the text.
R. The following sentence has been added: “Kinesiophobia levels are associated to foot function (14), patients with high levels of kinesiophobia present greater functional problems in the foot and lower extremity (40). In the case of hallux valgus, moderate or severe deformities are associated with higher levels of kinesiophobia and pain (15).”
It was important to notice that TSK-11 did not relate with SCDAI (disease activity), but with pain and education. Unfortunatelly this can be better discussed.
R. The discussion has been modified addressing that topic. Please find the whole discussion in the manuscript.
Conclusion
You said: “…SDAI and Body Mass Index are all related to a high level of kinesiophobia” and in discussion “ … A clear and significant tendency was not shown between a greater or lower level of kinesiophobia and the disease activity”. This is contradictory. Please review.
R. This part of the conclusion has been deleted as a clear and significant tendency was not shown between a greater or lower level of kinesiophobia and the disease activity.

Reviewer 2 Report
Relationship between kinesiophobia and foot pain and foot function, and disease activity in patients with rheumatoid arthritis: A cross sectional study
Thank you for the opportunity to review this paper which investigated the relationship between kinesiophobia and foot problems in people with rheumatoid arthritis.
Please see some comment below:
Introduction
1. Seems like a reasonable justification for the study
Methods
1. The methods present some results. Under the participants section, how the number of participants were recruited and what happened to those excluded should be in the results.
2. What did the interviews yield? Much of the findings could be questionnaire based. So, what was physical examination and what was written. Some of the data presented is not accounted for in the methods e.g. BMI
3. When were these people recruited? Was this from an existing clinic?
4. Some text may need adjusting. Are the fractions required? In this text from 2.4? The TSK-11 presents good coherence (a 1⁄4 0.68– 109 0.80), reliability test-retest (intraclass correlation coefficient [CCI] 1⁄4 0.72)
5. The sample size calculation needs further explanation. What effect was examined?
6. Some of the variables in Table 1 have relatively high SDs, are you certain all continuous variables were normally distributed?
7. Is there a threshold for kinesiophobia?
Results
1. The selection of variables to correlate with TSK-11 needs more explanation. How was educational background handled? This could be an ordinal variable, is it suitable for Pearson’s? The same goes for Table 3.
2. Use of ‘predictor’ should be tempered in a cross-sectional analysis. At best it is associated.
3. Reference to multivariable and multivariate is also used. Multivariable is preferred
Discussion
1. The conclusion does not match the results. Only Feet VAS and Educational background (which has some problems mentioned above) were significant in the model. Pearson’s correlations are inferior to multivariable models.
Minor:
Table 1
1. It would be easier to read with ‘Mean (SD)’ and ‘95% CI’ as the columns
2. IC should be CI. Both CI and Sd need defining in the footer, along with any abbreviations in the table
Author Response
Thank you for the opportunity to review this paper which investigated the relationship between kinesiophobia and foot problems in people with rheumatoid arthritis.
Introduction
- Seems like a reasonable justification for the study
R. Dear reviewer,
Thank you very much for giving us the possibility of addressing all the questions that arose during the review process. We think those comments have greatly improved the quality of this cross-sectional study. Please find below all the responses in a point-by-point fashion. In the new revised version, the changes are highlighted in red font.
Methods
- The methods present some results. Under the participants section, how the number of participants were recruited and what happened to those excluded should be in the results.
R. The participants´ results have been deleted from the methods section and they are now available in the results section.
- What did the interviews yield? Much of the findings could be questionnaire based. So, what was physical examination and what was written. Some of the data presented is not accounted for in the methods e.g. BMI
R. The interviews were carried out to achieve a better understanding of their responses, such as deeper explanations of their treatments. The following sentence has been added: “Also, the participants weight and height were assessed”.
- When were these people recruited? Was this from an existing clinic?
R. The following information has been added to the manuscript: “Participants were recruited from Hospital Universitario Virgen de las Nieves de Granada, in Spain, from the rheumatologist department. Participants were recruited from June 2021 to December 2021.”
- Some text may need adjusting. Are the fractions required? In this text from 2.4? The TSK-11 presents good coherence (a 1⁄4 0.68– 109 0.80), reliability test-retest (intraclass correlation coefficient [CCI] 1⁄4 0.72)
R. The fractions have been deleted to avoid confusion.
- The sample size calculation needs further explanation. What effect was examined?
R. The sample size was determined with EPIDAT 4.1 calculating an effect size of 0.61. This was calculated from a pilot study to evaluate the statistical power. Finally, the pilot study involved a sample of 30 subjects. The study was designed to detect changes exceeding 0.8 (high effect size) for a variation of the sample according to the above classification, with a type I error of 0.05 and a type II error of 0.2
- Some of the variables in Table 1 have relatively high SDs, are you certain all continuous variables were normally distributed?
R. Yes, only the age was not normally distributed.
- Is there a threshold for kinesiophobia?
R. TSK-11 does not permit different categories (small, medium or large), as it is a continuous variable.
Results
- The selection of variables to correlate with TSK-11 needs more explanation. How was educational background handled? This could be an ordinal variable, is it suitable for Pearson’s? The same goes for Table 3.
R. We totally agree, we have modified the variables and data, excluding educational background from the table.
- Use of ‘predictor’ should be tempered in a cross-sectional analysis. At best it is associated.
R. We have modified the information in the manuscript.
- Reference to multivariable and multivariate is also used. Multivariable is preferred
R. We totally agree, we have modified it.
Discussion
- The conclusion does not match the results. Only Feet VAS and Educational background (which has some problems mentioned above) were significant in the model. Pearson’s correlations are inferior to multivariable models.
R. The conclusion has been modified and “activity level SDAI” has been deleted from the conclusion, due to a clear and significant tendency was not shown between a greater or lower level of kinesiophobia and the disease activity: “The years of age, general VAS, feet VAS, education background, FFI physical activity and Body Mass Index are all related to a high level of kinesiophobia in patients with RA…”
Minor:
Table 1
- It would be easier to read with ‘Mean (SD)’ and ‘95% CI’ as the columns
R. The terms have been modified, and also the columns.
- IC should be CI. Both CI and Sd need defining in the footer, along with any abbreviations in the table
R. The term has been modified. Both CI and SD definition are now available in the footer: “SD: Standard deviation; CI: confidence interval”.

Reviewer 3 Report
1. A survey evaluation study. Evaluation methods for the hypothesis should be explained in more detail. It should be detailed how adequate the evaluations are for the severity of the disease and pain.
2. Perhaps the gender correlation could have been added.
The difference of the third study from other studies should be better explained in the discussion.
4. The results do not provide a new conclusion on this issue.
5. Limitations should be reviewed.
6. The discussion should be more descriptive and detailed according to existing publications.
Author Response
Dear reviewer,
Thank you very much for giving us the possibility of addressing all the questions that arose during the review process. We think those comments have greatly improved the quality of this cross-sectional study. Please find below all the responses in a point-by-point fashion. In the new revised version, the changes are highlighted in red font.
- A survey evaluation study. Evaluation methods for the hypothesis should be explained in more detail. It should be detailed how adequate the evaluations are for the severity of the disease and pain.
R. The justification of why we have performed the described cross-sectional study is because there was a lack of evidence about the role of foot pain, foot function, level of physical activity and educational background as a predictor of kinesiophobia in patients with RA.
- Perhaps the gender correlation could have been added.
R. As 73.4% of the participants were female, it was not significant to add a stratification by gender.
The difference of the third study from other studies should be better explained in the discussion.
R. Sorry, but we have not been able to answer this comment because we couldn’t recognise a third study in our manuscript.
- The results do not provide a new conclusion on this issue.
R. The methodology and results have been reviewed and modified.
- Limitations should be reviewed.
R. The following limitation has been added: “Finally, the mean age could also be considered as a limitation of the present study. The mean age (59.44 years) could have biased the results due to older patients supposedly present concomitant pathologies, that have not been considered, and they may have influenced the results in terms of kinesiophobia.”
- The discussion should be more descriptive and detailed according to existing publications.
R. The discussion has been modified. Please find the whole new discussion in the manuscript.

Round 2
Reviewer 2 Report
Thank you for the opportunity to re-review this work.
I have concerns about the conclusions. To me the most meaningful analysis is the modelling (table 3), but the conclusions appear to use the findings from the Pearson's correlations. It is difficult to claim that variables are 'related' simply because they correlate. I would recommend focusing the results and conclusions far more on the findings from the multivariable model. BMI and VASf are two variables that are related to your outcome variable - this needs to be highlighted.
The sample size calculation also needs more thought. It remains unclear what variable you are powered on. What primary variable did you use to make that calculation.
Why have you added a one-way ANOVA to the methods? ANOVA check for differences, the results present correlations?
Small errors also exist e.g. Table 1 has CI 95%, it should be 95% CI.